# The role of molecular xenomonitoring for assessing the impact of Ivermectin, Diethylcarbamazine Citrate and Albendazole triple-drug treatment against lymphatic filariasis: A cross-sectional study in the Kenyan coastal region

Henry Kanyi[1,2*], David Odongo[2], Walter Jaoko[2], Katherine Gass[3], Benard Chieng[1], Sylvie Araka[1], Collins Okoyo[1,4], Sammy M. Njenga[1]

**1** Eastern and Southern Africa Centre of International Parasite Control, Kenya Medical Research Institute, Nairobi, Kenya, **2** Department of Medical Microbiology, University of Nairobi, Nairobi, Kenya, **3** Neglected Tropical Diseases Support Center, Task Force for Global Health, Atlanta, Georgia, United States of America, **4** Department of Epidemiology, Statistics and Informatics, Kenya Medical Research Institute, Nairobi, Kenya

* hkanyi@kemri.go.ke

## Abstract

Lymphatic Filariasis (LF) is a mosquito-borne disease primarily caused by the filarial parasite *Wuchereria bancrofti*. The triple-drug therapy of Ivermectin, Diethylcarbamazine Citrate and Albendazole (IDA) has been adopted to accelerate LF elimination of LF as a public health problem, particularly in areas where onchocerciasis is not co-endemic. In Kenya, IDA was administered to endemic communities in Jomvu sub-County (Mombasa) and Lamu County in 2018 and 2019. The impact of IDA was assessed using Molecular Xenomonitoring (MX), which involves detecting *W. bancrofti* DNA in mosquito pools via quantitative PCR (qPCR). MX surveys were conducted at baseline (2018) and at IDA impact survey 1 (IIS-1) in 2021 after two treatment rounds. Mosquitoes were collected from 30 randomly and 5–9 purposively selected villages per area. Traps used included CDC light, Gravid, and BioGents (BG) traps at baseline, and only CDC light and Gravid traps at IIS-1. Mosquitoes were sorted by genus and pooled in up to 25 per pool. DNA was extracted and analyzed by qPCR. Filarial DNA prevalence was estimated using PoolTools v0.1.4. Concurrently, human infection was assessed using filariasis test strips (FTS) to detect circulating filarial antigen (CFA). At baseline, 37,732 mosquitoes were collected: 0.07% *Anopheles*, 2.3% *Aedes*, and 97.6% *Culex*. Of 486 pools tested, 30 (0.26%) were positive for *W. bancrofti* DNA. At IIS-1, 71,634 mosquitoes were collected: 0.46% *Anopheles* and 99.5% *Culex*. Of 1,695 pools tested, 22 (0.06%) were positive. Significant reductions were observed in filarial DNA in mosquitoes (76.9%, p < 0.001) and human antigenemia (52.1%, p = 0.004) after two treatment rounds. This study

**Data availability statement:** We have made available all the data used to draw the conclusions outlined in the manuscript. The datasets used in the analyses can be accessed in the GitHub via the link: https://github.com/mancollo/Kanyi-LF-Datasets.

**Funding:** This work received financial support from the Bill & Melinda Gates Foundation through their support of the Coalition for Operational Research on Neglected Tropical Diseases (COR-NTD) grant. COR-NTD is funded at The Task Force for Global Health primarily by the Bill & Melinda Gates Foundation and the United States Agency for International Development (USAID). The funders had no role in study design, data collection and analysis, decision to publish, or preparation of the manuscript.

**Competing interests:** The authors have declared that no competing interests exist.

demonstrates a substantial decline in LF infection following two rounds of IDA. It also highlights molecular xenomonitoring as a sensitive and non-invasive tool for monitoring LF elimination progress. We recommend its integration into national LF control programs as a complementary surveillance method post-MDA.

## 1. Introduction

Lymphatic filariasis (LF) is a vector-borne parasitic disease belonging to a category of diseases known as Neglected Tropical Diseases (NTDs). The disease is caused by three filarial worms: namely, *Wuchereria bancrofti, Brugia malayi* and *B. timori*. The filarial parasites are transmitted by different types of mosquitoes, which vary from one geographical region to another. In Kenya, the infection is known to be transmitted by mosquitoes belonging to two genera: *Anopheles (Anopheles funestus* and *Anopheles gambiae*) and *Culex (Culex quinquefasciatus*) [1–4]. The disease is primarily confined to the coastal region in Kenya where ecological factors are most suitable for its transmission [5]. Currently, about 4 million people are estimated to be at risk of infection in the Kenyan coastal region [4,6].

The World Health Organization (WHO) established the Global Programme for Elimination Lymphatic Filariasis (GPELF) in the year 2000 in response to the World Health Assembly resolution WHA50.29 formulated in 1997. The resolution urged Member States to immediately initiate activities to eliminate LF as a public health problem by the year 2020. The GPELF has the mandate to steer efforts toward the interruption of LF transmission, and morbidity management and disability prevention [7]. To interrupt transmission of disease, the GPELF recommends annual community-wide mass drug administration (MDA) of antifilarial tablets to all eligible individuals living in the endemic areas for a minimum of five rounds and with treatment coverage of above 65% [7–9]. According to WHO, a total of 9.7 billion cumulative treatments were delivered to more than 943 million people at least once in 71 countries from 2000 to 2023 with a total of 412.4 million people being reported to have been treated in 32 countries in 2023 [7].

The Kenyan National Programme for Elimination of Lymphatic Filariasis (NPELF) was launched in 2002 and mainly used a two-drug regimen (diethylcarbamazine and albendazole) for mass drug administration (MDA). The Programme, initially focused on Kilifi but was expanded in 2003 to include Lamu, Tana River, and Kwale [10]. However, the MDA campaigns from 2002 to 2011 were not given every year and did not attain the required treatment coverage as recommended by WHO. This necessitated restarting of the MDA campaigns in 2015 [7,11]. The programme was further extended to Mombasa and Taita Taveta in 2015 and 2016, respectively [12]. A study conducted prior to the MDA in 2015 reported a 1.3% mean prevalence of LF in coastal Kenya, with the highest rate of 6.3% observed in Lamu County [13].

Previous small-scale clinical studies have indicated that triple-drug therapy consisting of Ivermectin, Diethylcarbamazine Citrate (DEC) and Albendazole referred to IDA is capable of clearing microfilariae (Mf) in infected persons and its safe for

human use [14]. As a result, WHO allowed and provided an updated treatment guideline that endorsed the use of IDA as an alternative MDA therapy for LF elimination programmes in November 2017 [15]. In Kenya, Lamu County and Jomvu Sub-County in Mombasa County were selected as two sites where MDA with IDA was to be conducted to accelerate the elimination of LF based on the existing LF prevalence data and endemicity [13]. An operational research study was also designed for monitoring and evaluation of IDA MDA and identifying appropriate indicator(s) to be used for assessing the impact of the triple-drug therapy in future.

Molecular xenomonitoring (MX) has been recommended as a sensitive complementary method for the evaluation of LF in the endemic countries using molecular based quantitative real-time polymerase chain reaction (qPCR) assay. The qPCR assay has been shown to be sensitive and specific compared to conventional PCR in detecting the *W. bancrofti* parasite [16]. In a qPCR assay, a positive sample is detected by accumulation of DNA amplification fluorescent signal denoted as cycle threshold (Ct) value. Ordinarily, a sample with higher concentration of target DNA has lower Ct levels since amplification is detected during the early cycles of the assay. Due to the fact that qPCR Ct values significantly (inversely) correlate with filarial parasite DNA, this method can provide information on intensity of infection in the mosquito vector [16,17].

Since MX utilizes mosquito samples in detection of filarial DNA as a proxy indicator for ongoing LF transmission, it therefore offers an alternative to the use of human blood making it both less ethically demanding and minimally invasive and intrusive to human [18]. The tool is additionally advantageous since the collection of mosquitos is less laborious and requires no cold chain for storage in the field and during transportation of the samples to the laboratory [19]. Moreover, MX is a high throughput sensitive technique thus making it a practical choice for large-scale surveillance efforts for the detection of any residual infection and/or any ongoing transmission accurately. This study therefore utilized MX to assess the presence of filarial DNA in mosquito vectors in the LF endemic communities living in Jomvu sub-County, Mombasa County and Lamu County in the Kenyan Coast. These data, other than informing next steps to the control programme, will also support in the development of a dossier to support application to WHO by the Kenyan Ministry of Health for recognition to have eliminated LF as a public health problem in the country.

## 2. Materials and methods

### 2.1. Ethics statement

The study received ethical approval from the Scientific and Ethics Review Unit (SERU) of Kenya Medical Research Institute (KEMRI) (KEMRI/SERU protocol No. 4523). Permission for human and mosquito sample collection was further sought and obtained from the respective county governments. Community mobilization and sensitization was done through meetings with the local leaders and community members at the village/cluster level. All adult participants provided individual written informed consent. For participants under 18, written parental permission was obtained, along with written assent from the minor where appropriate. The consent process, conducted in the local language, included provisions for non-literate participants and specified that samples could be stored for future molecular analysis related to vector-borne disease research.

### 2.2. Study design

This study utilized samples that were collected in repeated cross-sectional entomological surveys that were carried out in the two evaluation units (EU), namely Lamu County and Jomvu sub-County in Mombasa County. In each EU, samples were collected from 30 randomly and 5–9 purposively selected villages also referred to as clusters at baseline and at IIS-1. Random clusters were selected purely by chance and this was done using an excel selection tool. Purposive clusters were selected intentionally based on suspicion of high LF prevalence and in consultation with national and local authorities.

## 2.3. Study site

The mosquito samples were collected in the same sites (Lamu East, Lamu West, and Jomvu sub-Counties) as the population-based surveys as described by Njenga *et al.,* [20]. Lamu East and Lamu West sub-Counties situated in Lamu County were considered to be a single EU due to their low population while Jomvu sub-County in Mombasa County was the second EU. Lamu County consists mainly of the mainland and several islands in the Indian Ocean which extend to the Northern part of the Kenyan coast and border Somalia. Jomvu sub-County is located to the west of Mombasa city and characterized by urban, peri-urban and rural areas.

## 2.4. Community human surveys

The human surveys were also done in the same villages as mosquito surveys. The sample collection for baseline survey was conducted from the 22nd October – 10th November 2018 while IIS-1 survey was conducted from the 12th April – 7th May 2021. In the field, consent for the enrolment of all the participants was done on an opt-in basis and further, only children whose parents and/or legal guardians gave informed consent were recruited. Testing for LF was done using Filariasis Test Strip (FTS; Alere, Scarborough, ME) as per the manufacturer's instructions and standard operating procedures (SOPs) [21,22]. A finger prick sample of blood was collected by an experienced and trained phlebotomist into a capillary tube calibrated to collect 75µl. The sample was added onto a test pad of the FTS placed on a flat-levelled surface and away from direct sunlight. The strip was allowed to run and read at exactly 10 minutes. From participants whose samples turned positive, an additional blood sample was collected during the peak hours of Mf circulation (10.00pm - midnight) and used to prepare a thick blood smear and for microscopic examination. Further, another dried blood spot (DBS) sample was collected from the FTS positive individual for further PCR based analysis alongside the mosquito samples. The samples were later transported to the KEMRI/ESACIPAC NTD research laboratory in Nairobi and stored at -20°C until laboratory analysis was done.

## 2.5. Study population

The samples were collected from two categories of human populations as follows: - children from 5 - 9 years and an older population (age ≥ 10 years in baseline and age ≥ 18 years at IIS-1). The upper age group was changed to ≥ 18 years for the impact assessment because it became clear that the 10–17 years age group would be challenging for NTD programs to assess and interpret if it were to become global guidance, due to varying rates of boarding school attendance and the age at which children enter the work sector. The sample size of the 5–9 years-old children that was required for the survey in each area was selected to be similar to the sample sizes recommended for the WHO-recommended TAS survey.

## 2.6. Mosquito sampling and pooling

Mosquito samples were collected from villages, also referred to as clusters, and which served as the primary sampling unit (PSU). The samples were collected in 6 households per cluster at baseline (before treatment) from the 14th October – 18th November 2018. At IIS-1 survey, which was done after two annual rounds of IDA MDA, mosquito sample collection was done in 5 households per cluster and in two rounds: round one was done from 24th November 2020 – 27th January 2021 while round 2 collection was done from 7th – 18th July 2021. Mosquitoes were collected using CDC light traps, Gravid traps and BioGents (BG) sentinel traps at baseline while at IIS-1 only CDC light and Gravid traps were used [1]. In the field, mosquitoes were identified morphologically and sorted using the morphological keys [23–25]. Mosquitoes were then pooled with each pool containing a maximum of 25 individual mosquitoes. During mosquito's collection, efforts were made to ensure as many complete pools as possible were formed to enhance power of the data and the remaining mosquitoes put in a separate smaller pool [26,27]. This was also meant to maximize the number of complete pools formed for analysis to be done at the pool-level. The samples were later transported to a KEMRI/ESACIPAC NTD research laboratory where analysis using qPCR assay was performed.

## 2.7. Selection of the mosquito pools

During baseline assessment, large numbers of *Culex* mosquitoes were collected, while the numbers of *Anopheles* and *Aedes* mosquitoes were relatively small. As a result of the above, during IIS-1 the collection of *Anopheles* was of paramount interest to the study since it is considered to be a primary vector for LF in the region and thus highly prioritized. Additionally, BG Trap which mainly targets the *Aedes* was dropped based on baseline qPCR results and available literature that does not include it as a primary LF vector in Kenya [1]. To enable maximum yield of the mosquitoes, the collection during the IIS-1 was done in two rounds (round 1 & 2). The timings were largely determined and dependent on the rainy season(s), which is the period when mosquitoes breeding is at its peak.

## 2.8. Filarial DNA extraction from mosquito pools and night blood dried blood spot samples

Total Genomic DNA (gDNA) was extracted from mosquitoes and night blood DBS samples using the Qiagen DNeasy extraction kit (Qiagen, Valencia, CA) as per the manufacturer's instructions and the relevant standard operating procedures (SOPs) [27]. To verify the consistency and success of DNA extraction procedure, 1µl of an internal amplification control (IAC) from pDMD801 plasmid at a concentration of 100pg/µl was added to the sample [28,29]. The extracted sample was double eluted using 100ul of elution buffer (AE). The final eluate/sample was stored at 4°C temporarily until qPCR was done and transferred to – 20°C for long term storage. A negative extract control was used as a template for quality control.

## 2.9. Real-time PCR assay for the detection of *Wuchereria bancrofti* DNA

Real-time PCR (qPCR) assays were performed on the StepOnePlus Real-Time PCR System (Applied Biosystems) for the detection of *W. bancrofti* DNA (Wb-Cl1 assay). The samples were run in duplicate wells in an optical 96-well reaction plate using TaqPath ProAmp Master Mix; (Applied Biosystems), as previously described by Zulch *et al.,* [30]. The primers sequences were as follows, the forward primer was 5′- GCTGAAAAACATTCGCTTTTGAATG-3′, the reverse primer 5′-GGGTAATTAAACCGGTGATCCT-3′, and the probe was 5′-/**56-FAM**/ACAACAACT/**ZEN**/ATATGGGAATGGTG-CAGGT/**3IABKFQ**/-3′. The assay's cycling conditions were as follows: - a 2-minute initial hold at 50°C and a 10-minute incubation period at 95°C. Incubation period was followed by 40 cycles of 95°C for 15 seconds and 60°C for 1 minute for denaturation of template and annealing/extension, respectively.

An internal amplification control (IAC) was also run per sample prior to the *W. bancrofti* assay to test for successful DNA extraction and to ensure exclusion of any presence of qPCR inhibitors. Amplification of internal control (pDMD801 plasmid) was performed in a 7µl reaction mixture containing qPCR buffer (TaqPath ProAmp Master Mix, Applied Biosystems), 12.5 pmol of each pDMD801 plasmid primer (F: 5′-CTAACCTTCGTGATGAGCAATCG -3′, R: 5′- GATCAGC-TACGTGAGGTCCTAC -3′, 2.5pmol of the pDMD801 plasmid double-labeled probe: 5′-/**56FAM**/AGCTAGTCG/**ZEN**/ATG-CACTCCAGTCCTCCT/**3IABkFQ**/ -3′), and 2µl of the DNA sample. The IAC assay cycling conditions were as follows: - a 2-minute incubation step at 50°C and 10-minute incubation at 95°C. This was followed by 40 cycles, each of 15 seconds at 95°C, 30 seconds at 59°C, and 30 seconds at 72°C [29]. Preparation of master mix and addition of DNA template were done in separate laminar flow chambers which had previously been sterilized using ultraviolet (UV) light for 10 minutes and subsequently by freshly made 10% bleach. Each plate done in the qPCR assay had both positive and negative controls included for quality control and quality assurance purposes.

## 2.10. Data management and analysis

During the field survey, data was collected on smartphone devices and continuously uploaded to an electronic central database using the open source software Secure Data Kit (SDK). Field staff were trained on using the tools for data collection in the field. QR code stickers were used to protect participant identities and label all specimens. The data were

finally stored in a SQL secure data server. All data resulting from laboratory testing were entered into password protected computer and exported into an Excel sheet. The data was stored in a secure database and ultimately linked with the individual level questionnaire and field results by the unique QR code.

The analysis for the means and distributions across villages, study areas/EUs, and genus was calculated using STATA version 18.5. Filarial DNA prevalence in mosquitoes was computed using software package PoolTools v0.1.4 [31]. Since PoolTools v0.1.4 does not take survey design into account, the upper 1-sided 95% confidence interval was calculated assuming a hypergeometric distribution using STATA v18.5 to account for the survey design. To assess the impact of IDA, the relative reductions (RR) in infection prevalence at the impact survey compared to baseline survey were calculated using multivariable mixed effect models with random intercepts for the villages and evaluation units and p-values obtained using Wald test. The relative reduction was calculated using the following formula

$$RR = \left[ \frac{Baseline\ prevalence - Impact\ prevalence}{Baseline\ prevalence} \right] x\ 100\%.$$

## 3. Results

### 3.1. General characteristics

During the baseline survey, mosquitoes were collected in 210 households spread across the 35 villages in each EU. Overall, 30,799 female mosquitoes were collected in Jomvu and 6,933 collected in Lamu. Genus *Culex* had the largest number of the mosquitoes collected with 30,157 and 6,680 in Jomvu and Lamu, respectively. The distribution of the mosquitoes collected during baseline assessment was 25 (0.07%), 870 (2.30%) and 36,837 (97.63%) for *Anopheles, Aedes* and *Culex,* respectively. The mosquitoes were sorted into 323 and 163 pools of up to 25 mosquitoes in Jomvu and Lamu, respectively (Table 1).

During IIS-1, the collection of mosquitoes was done in two rounds to increase the number of *Anopheles* genus. In each round, 750 households were sampled across 74 clusters (39 in Jomvu and 36 in Lamu). A total of 54,702 and 16,932 female mosquitoes were collected in Jomvu and Lamu, respectively. These mosquitoes consisted of 332 (0.46%) and 71,302 (99.54%) for genus *Anopheles* and *Culex*, respectively. As was the case in the baseline survey, at IIS-1, genus

**Table 1. Summary of mosquitoes collected in all traps by the evaluation unit and village type during baseline (2018) and impact (2021) surveys in Jomvu and Lamu.**

| Survey | EU | Village type (n: number of villages) | Number of *Anopheles* mosquitoes collected median (range); n | | Number of *Culex* mosquitoes collected median (range); n | | Number of *Aedes* mosquitoes collected median (range); n | |
|---|---|---|---|---|---|---|---|---|
| | | | Males | Females | Males | Females | Males | Females |
| Baseline (2018) | Jomvu | Random (n=30) | 0 (0–1); n=1 | 0 (0–2); n=17 | 12 (0–196); n=3467 | 112 (0–875); n=25566 | 0 (0–98); n=287 | 1 (0–83); n=540 |
| | | Purposive (n=5) | 0 (0–1); n=1 | 0 (0–2); n=4 | 11.5 (0–91); n=588 | 102 (0–525); n=4591 | 0 (0–6); n=18 | 0 (0–24); n=81 |
| | | All sites (n=35) | 0 (0–1); n=2 | 0 (0–2); n=21 | 12 (0–196); n=4055 | 110 (0–875); n=30157 | 0 (0–98); n=305 | 1 (0–83); n=621 |
| | Lamu | Random(n=30) | 0 | 0 (0–1); n=3 | 0 (0–256); n=1213 | 9 (0–350); n=5911 | 0 (0–6); n=28 | 0 (0–15); n=119 |
| | | Purposive(n=5) | 0 | 0 (0–1); n=1 | 1 (0–57); n=152 | 21 (0–100); n=769 | 0 (0–6); n=23 | 0 (0–34); n=130 |
| | | All sites(n=35) | 0 | 0 (0–1); n=4 | 1 (0–256); n=1365 | 11 (0–350); n=6680 | 0 (0–6); n=51 | 0 (0–34); n=249 |
| Impact (2021) | Jomvu | Random(n=30) | 0 (0–36); n=49 | 0 (0–70); n=126 | 7 (0–560); n=7244 | 74 (0–775); n=41593 | 0 | 0 |
| | | Purposive(n=9) | 0 (0–4); n=13 | 0 (0–41); n=158 | 9 (0–185); n=2386 | 91 (0–515); n=12825 | 0 | 0 |
| | | All sites(n=39) | 0 (0–36); n=62 | 0 (0–70); n=284 | 8 (0–560); n=9630 | 75 (0–775); n=54418 | 0 | 0 |
| | Lamu | Random(n=30) | 0 (0–11); n=28 | 0 (0–10); n=48 | 3 (0–450); n=2892 | 14 (0–400; n=13612 | 0 | 0 |
| | | Purposive(n=6) | 0 (0–7); n=7 | 0 | 4 (0–190); n=1051 | 35 (0–545); n=3272 | 0 | 0 |
| | | All sites (n=36) | 0 (0–11); n=35 | 0 (0–10); n=48 | 3 (0–450); n=3943 | 17 (0–545); n=16884 | 0 | 0 |

*Culex* had the largest number of mosquitoes with 54,418 and 16,884 in Jomvu and Lamu, respectively. Similarly, Genus *Culex* had the highest number of analysed pools with 1,630 while *Anopheles* had 65 pools. Across the two EUs, the random villages had higher numbers of the mosquitoes collected than the purposive villages. The number of mosquitoes collected per trap is as shown in Table 1.

### 3.2. Mosquito collection by trap types

The number of mosquitoes collected varied by trap types in both Jomvu and Lamu. The Gravid trap had the highest number of collections with 28,953 mosquitoes at the baseline survey and 43,260 mosquitoes at the IIS-1 survey. The light trap yielded the second greatest number of mosquitoes with 7,628 and 28,374 at baseline and IIS-1, respectively. The BG-sentinel trap which was only deployed during the baseline survey collected a total of 1,151 mosquitoes (Fig 1).

### 3.3. Filarial DNA prevalence in mosquitoes by evaluation units and village type

At baseline, Jomvu had the highest prevalence of filarial DNA at 0.40% (95% CI: 0.27-0.56), followed by Lamu with 0.00% (95% CI: 0.00-0.05). At IIS-1, Jomvu and Lamu had prevalence of 0.09% (95% CI: 0.06-0.14) and 0.01% (95% CI: 0.00-0.04), respectively. At baseline, both the random and purposive villages had almost similar prevalence in both Jomvu and Lamu. At IIS-1, the prevalence was 0.09% (95% CI: 0.05-0.14) for random villages and 0.10% (95% CI: 0.04-0.19) for purposive villages in Jomvu. In Lamu, the prevalence was 0.01% (95% CI: 0.00-0.03) for random villages and 0.04% (95% CI: 0.00-0.16) for purposive villages (Table 2).

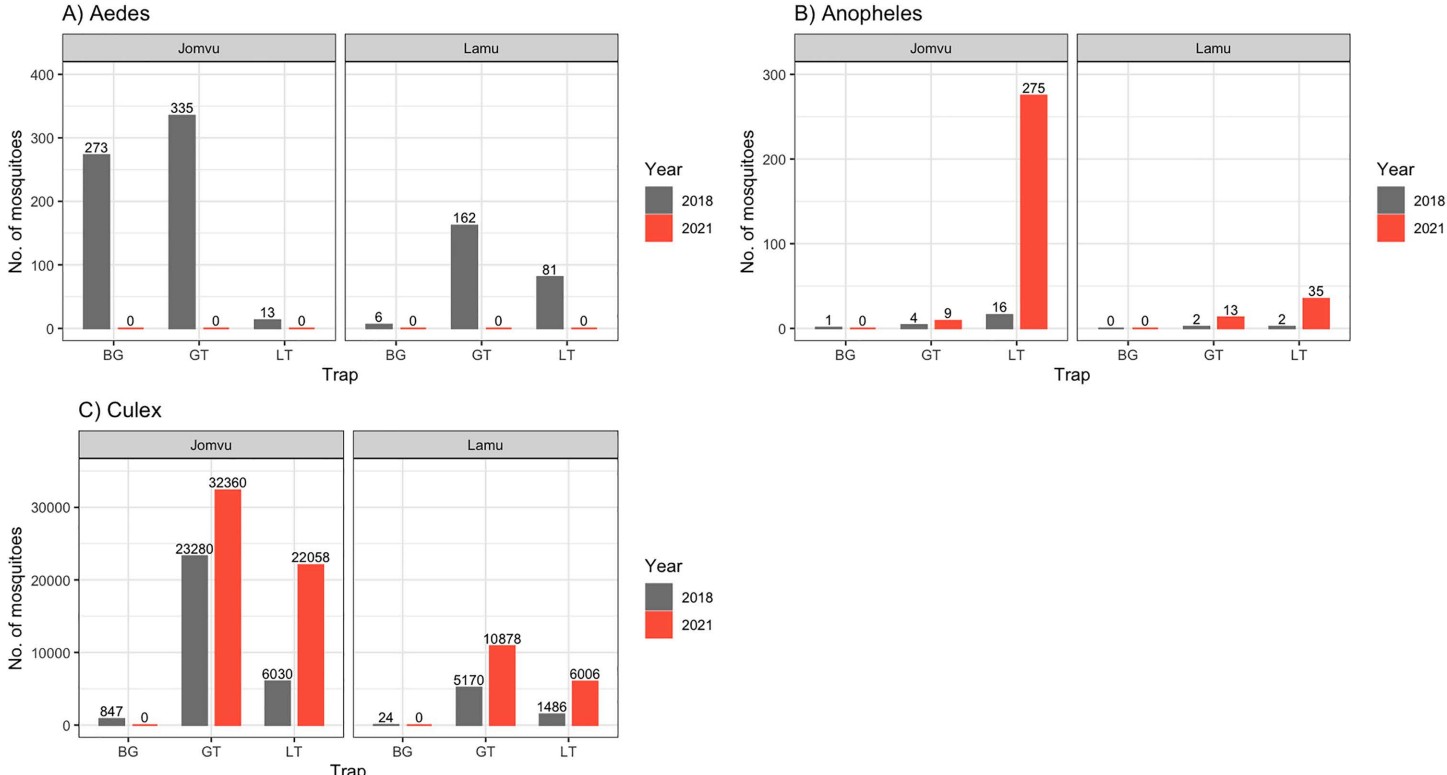

**Fig 1. The total number of female mosquitoes collected using BioGents sentinel trap (BG), gravid trap (GT), and light trap (LT) during baseline (2018) and IDA Impact Survey -1 (IIS-1: 2021) in Jomvu and Lamu evaluation units.**

**Table 2. Estimated prevalence of filarial DNA in mosquitoes % (95% CI) by village type using real time qPCR technique for baseline (2018) and impact (2021) surveys in Jomvu and Lamu evaluation units.**

| Village Type | Evaluation Units (EU) | | | | | | | | |
|---|---|---|---|---|---|---|---|---|---|
| | Jomvu | | | Lamu | | | Overall | | |
| **Prevalence Estimates (Prev) at baseline, 2018** | | | | | | | | | |
| | No. of pools analyzed | No. of positive pools | Prevalence % (95%CI) | No. of pools analyzed | No. of positive pools | Prevalence % (95%CI) | No. of pools analyzed | No. of positive pools | Prevalence % (95%CI) |
| Random (n=60) | 279 | 26 | 0.40 (0.27-0.57) | 133 | 0 | 0 (0.00-0.06) | 412 | 26 | 0.26 (0.18-0.38) |
| Purposive (n=10) | 44 | 4 | 0.39 (0.12-0.89) | 30 | 0 | 0 (0.00-0.26) | 74 | 4 | 0.22 (0.07-0.52) |
| All sites (n=70) | 323 | 30 | **0.40 (0.27-0.56)** | 163 | 0 | **0.00 (0.00-0.05)** | 486 | 30 | **0.26 (0.18-0.36)** |
| **Prevalence Estimates (Prev) at IIS-1, 2021** | | | | | | | | | |
| Random (n=60) | 661 | 13 | 0.09 (0.05-0.14) | 587 | 1 | 0.01 (0.00-0.03) | 1248 | 14 | 0.05 (0.03-0.08) |
| Purposive (n=15) | 332 | 7 | 0.10 (0.04-0.19) | 115 | 1 | 0.04 (0.00-0.16) | 447 | 8 | 0.08 (0.04-0.15) |
| All sites (n=75) | 993 | 20 | **0.09 (0.06-0.14)** | 702 | 2 | **0.01 (0.00-0.04)** | 1695 | 22 | **0.06 (0.04-0.08)** |
| **Relative reduction (RR)** | | | | | | | | | |
| Random | RR=77.5, p<0.001* | | | Increase | | | RR=80.8, p<0.001* | | |
| Purposive | RR=74.4, p=0.016* | | | Increase | | | RR=63.6, p=0.065 | | |
| All sites | RR=77.5, p<0.001* | | | Increase | | | RR=76.9, p<0.001* | | |

Prevalence Estimates computed using software package PoolTools v0.1.4.

* Indicate statistical significance (p < 0.05).

RR: relative reduction.

n: **number of villages**.

Assessment of the prevalence reductions between baseline and IIS-1 surveys showed that there was significant relative reduction in prevalence in random (RR=80.8, p<0.001) and in all sites (overall) RR=76.9, p<0.001 but not in purposive sites (RR=63.6, p=0.065) in Jomvu. However, in Lamu there was increase in prevalence in both random and purposive villages (Table 2).

### 3.4. Prevalence of filarial DNA by genus

Overall, only *Culex* mosquitoes were found to habour detectable amounts of filarial DNA by the qPCR method at 0.26% (95% CI: 0.18-0.38) during baseline and 0.06% (95% CI: 0.04-0.08) at IIS-1. Jomvu had higher prevalence of filarial DNA at 0.40% (95% CI: 0.27-0.56) during baseline and 0.09% (95% CI: 0.06-0.14) during IIS-1. There was no filarial DNA detected in either the *Anopheles* and *Aedes* mosquitoes at both baseline and IIS-1 (Table 3).

Jomvu had a significant relative reduction in filarial DNA prevalence of RR=78.0%, p<0.001 for *Culex* mosquitoes. However, results for Lamu showed an increase in filarial DNA prevalence (Table 3).

### 3.5. Comparison of filarial DNA prevalence in mosquitoes and circulating filarial antigen in human population results

We compared the prevalence of filarial DNA in the mosquitoes against CFA in humans as shown in Table 4. At baseline, the filarial DNA prevalence was 0.26% (CI: 0.18-0.38) while at the IIS-1 it was 0.06% (CI: 0.04-0.08). The comparison showed that there was a significant relative reduction in filarial DNA prevalence (RR=76.9%, p<0.001) from baseline to IIS-1. Comparatively, a similar trend was observed for the CFA results that showed prevalence of 0.94% (CI: 0.73 -1.21) at baseline to 0.45% (CI: 0.34-0.60) at IIS-1. Similarly, a significant relative reduction of RR=52.11%, p=0.004* was observed between baseline and IIS-1 by human's CFA. In both the filarial DNA and human CFA prevalence, there were

**Table 3. Estimated prevalence of filarial DNA % (95%CI) by mosquito genus using real time qPCR technique for baseline (2018) and impact (2021) surveys in Jomvu and Lamu evaluation units.**

| Survey | Village type | Female *Culex* | | | Female *Anopheles* | | | Female *Aedes* | | |
|---|---|---|---|---|---|---|---|---|---|---|
| | | Prevalence % (95%CI) | No. of pools | No. of positive pools | Prevalence % (95%CI) | No. of pools | No. of positive pools | Prevalence % (95%CI) | No. of pools | No. of positive pools |
| **Baseline (2018)** | | | | | | | | | | |
| Overall | Random (n=60) | 0.26 (0.18-0.38) | 399 | 26 | 0.00 | 0 | 0 | 0.00 (0.00–1.10) | 13 | 0 |
| | Purposive (n=10) | 0.22 (0.07-0.52) | 73 | 4 | 0.00 | 0 | 0 | 0.00 (0.00–14.79) | 1 | 0 |
| | All sites (n=70) | 0.26 (0.18-0.37) | 472 | 30 | 0.00 | 0 | 0 | 0.00 (0.00–1.03) | 14 | 0 |
| Jomvu | Random (n=30) | 0.41 (0.27-0.59) | 267 | 26 | 0.00 | 0 | 0 | 0 (0.00–1.28) | 12 | 0 |
| | Purposive (n=5) | 0.39 (0.12-0.90) | 43 | 4 | 0.00 | 0 | 0 | 0.00 (0.00–15) | 1 | 0 |
| | All sites (n=35) | 0.40 (0.27-0.56) | 310 | 30 | 0.00 | 0 | 0 | 0.00 (0.00–11.90) | 13 | 0 |
| Lamu | Random (n=30) | 0.00 (0.00-0.01) | 132 | 0 | 0.00 | 0 | 0 | 0.00 | 0 | 0 |
| | Purposive (n=5) | 0.00 (0.00-0.26) | 30 | 0 | 0.00 | 0 | 0 | 0.00 (0.00–7.40) | 1 | 0 |
| | All sites (n=35) | 0.00 (0.00-0.05) | 163 | 0 | 0.00 | 0 | 0 | 0.00 (0.00–7.40) | 1 | 0 |
| **IIS-1(2021)** | | | | | | | | | | |
| Overall | Random (n=60) | 0.05 (0.03-0.08) | 1214 | 14 | 0.00 (0.00–0.97) | 34 | 0 | | | |
| | Purposive (n=15) | 0.08 (0.04-0.16) | 416 | 8 | 0.00 (0.00–1.0) | 31 | 0 | | | |
| | All sites (n=75) | 0.06 (0.04-0.08) | 1630 | 22 | 0.00 (0.00–0.49) | 65 | 0 | | | |
| Jomvu | Random (n=30) | 0.09 (0.05-0.15) | 630 | 13 | 0.00 (0.00–1.02) | 31 | 0 | | | |
| | Purposive (n=9) | 0.1 (0.04-0.20) | 301 | 7 | 0.00 (0.00–1.00) | 31 | 0 | | | |
| | All sites (n=39) | 0.09 (0.06-0.14) | 931 | 20 | 0.00 (0.00–0.52) | 62 | 0 | | | |
| Lamu | Random (n=30) | 0.01 (0.00-0.03) | 584 | 1 | 0.00 (0.00–16.02) | 3 | 0 | | | |
| | Purposive (n=6) | 0.04 (0.00-0.16) | 115 | 1 | 0.00 | 0 | 0 | | | |
| | All sites (n=36) | 0.01 (0.00-0.04) | 699 | 2 | 0.00 (0.00–16.02) | 3 | 0 | | | |
| **Relative reduction** | | | | | | | | | | |
| Overall | Random | RR=81.4, p<0.001* | – | – | 0.00 | – | – | | | |
| | Purposive | RR=65.2, p=0.080 | – | – | 0.00 | – | – | | | |
| | All sites | RR=76.9, p<0.001* | – | – | 0.00 | – | – | | | |
| Jomvu | Random | RR=78.0, p<0.001* | – | – | 0.00 | – | – | | | |
| | Purposive | RR=74.4, p=0.022* | – | – | 0.00 | – | – | | | |
| | All sites | RR=78.0, p<0.001* | – | – | 0.00 | – | – | | | |
| Lamu | Random | Increase | – | – | 0.00 | – | – | | | |
| | Purposive | Increase | – | – | 0.00 | – | – | | | |
| | All sites | Increase | – | – | 0.00 | – | – | | | |

Prevalence Estimates computed using software package PoolTools v0.1.4.

* Indicate statistical significance (p<0.05).

RR: relative reduction.

n: **number of villages**.

significant and non-significant relative reductions of the infection from baseline to IIS-1 surveys in the random and purposive villages, respectively. In Jomvu, one person was Mf positive at baseline by microscopic examination of a nighttime thick blood smear. At IIS-1, no positive individuals were identified by microscopy, while two individuals were identified as positive based on night blood DBS by qPCR; both cases were observed in random villages (Table 4). Additionally, there

**Table 4. Comparison between molecular and parasitological based indicators for the assessment of LF infection.**

| Survey | Village type | Molecular assessment | | Parasitological assessment | |
|---|---|---|---|---|---|
| | | Prevalence by any female mosquito | Prevalence in human by night blood sample | Prevalence in human by FTS test | Prevalence in human by Mf |
| Baseline (2018) | | | | | |
| Overall | Random (n=60) | 0.26 (0.18–0.38) | 0.00 | 0.84 (0.63–1.13) | 0.02 (0.00–0.13) |
| | Purposive (n=10) | 0.22 (0.07–0.52) | 0.00 | 1.47 (0.88–2.48) | 0.00 |
| | All sites (n=70) | 0.26 (0.18–0.36) | 0.00 | 0.94 (0.73–1.21) | 0.02 (0.00–0.11) |
| Impact (2021) | | | | | |
| Overall | Random (n=60) | 0.05 (0.03–0.08) | 0.02 (0.01–0.08) | 0.42 (0.30–0.57) | 0.00 |
| | Purposive (n=15) | 0.08 (0.04–0.15) | 0.00 | 0.69 (0.36–1.33) | 0.00 |
| | All sites (n=75) | 0.06 (0.04–0.08) | 0.02 (0.00–0.07) | 0.45 (0.34–0.60) | 0.00 |
| Relative Reduction Rates - Overall | Random | RR=80.8, p<0.001* | Increase | RR=50.84, P=0.015* | RR=100.00, P<0.001* |
| | Purposive | RR=63.6, p=0.065 | | RR=52.79, P=0.197 | 0.00 |
| | All sites | RR=76.9, p<0.001* | Increase | RR=52.11, p=0.004* | RR=100.00, P<0.001* |

Prevalence Estimates computed using software package PoolTools v0.1.4.

* Indicate statistical significance (p<0.05).

RR: relative reduction.

n: **number of villages**.

was a significant positive correlation between the filarial DNA and human CFA results except in Lamu at baseline survey (Table 5).

### 3.6. Distribution of the filarial DNA in mosquitoes across the two evaluation units

Although all the villages in Lamu had negative results for filarial DNA at baseline, at IIS-1 the EU had positive mosquito pools in two villages (Ndau and Wiyoni) with a prevalence of 0.2% each. In Jomvu, there was a decline in the prevalence from baseline to IIS-1; however, the two villages of Mikanjuni and Mwamlai had relatively higher prevalence at IIS-1, with 0.9% and 0.6%, respectively (Fig 2).

### 4. Discussion

This study has demonstrated a significant impact of IDA triple therapy on LF infection levels in Kenya through an overall LF prevalence reduction rate of 76.9% in the mosquito vectors. This was observed through a decline in prevalence from 0.26% at baseline to 0.06% at the IIS-1. A similar study done in India showed a significant drop in pool positivity

**Table 5. Correlation between mosquito positive pools and FTS positive individuals by evaluation unit and survey.**

| Survey | EU | Number of villages | Number of mosquito pools | Number of mosquito positive pools | Number of individuals tested | Number of FTS positive individuals | Correlation coefficient¶ positive pools and FTS individuals [r, p-value] | Kappa index (% agreement) |
|---|---|---|---|---|---|---|---|---|
| Baseline (2018) | Jomvu | 35 | 323 | 30 | 3398 | 45 | r=0.534, p<0.001* | k=0.72 (A=85.71%) |
| Baseline (2018) | Lamu | 35 | 163 | 0 | 3226 | 15 | r=0.000† | k=0.00 (A=74.29%) |
| Impact (2021) | Jomvu | 39 | 993 | 20 | 5459 | 36 | r=0.333, p=0.038* | k=0.59 (A=79.49%) |
| Impact (2021) | Lamu | 36 | 702 | 2 | 5230 | 12 | r=0.972, p<0.001* | k=0.30 (A=88.89%) |

¶The Pearson correlation coefficient was estimated using positivity data for all clusters.

†No correlation was observed in Lamu during baseline.

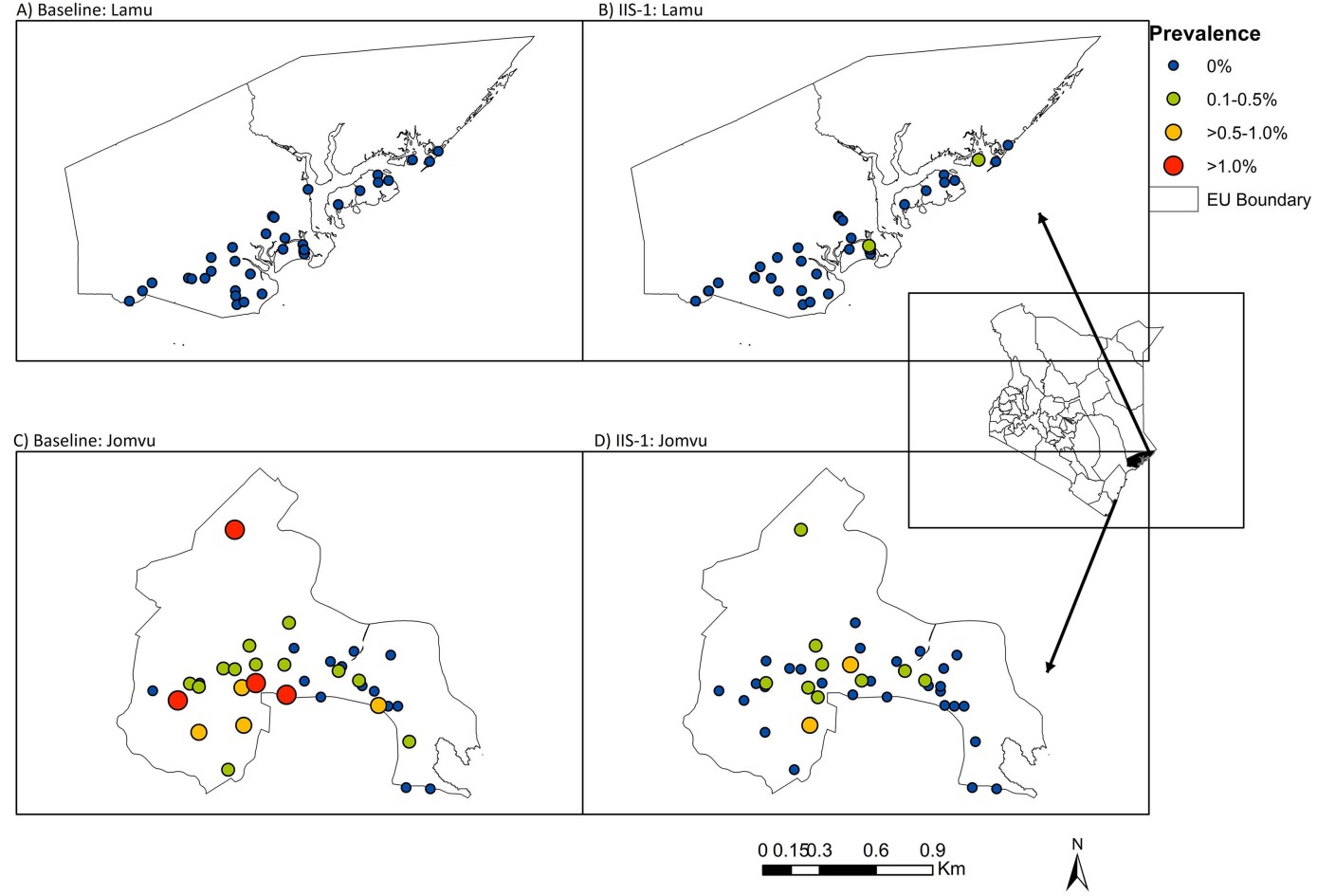

**Fig 2. A map showing the two evaluation units and overall filarial DNA prevalence in mosquitoes by villages at baseline and IDA Impact Survey 1 (IIS-1).** The inset map of Kenya shows the location of the two study areas. The map was created using ArcGIS Desktop version 10.2.2 software (Environmental Systems Research Institute Inc., Redlands, CA, USA). The base layer of the map was obtained from Environmental Systems Research, Inc. (ESRI) (https://www.esri.com/en-us/arcgis/products/arcgis-online/resources).

of mosquitoes at two time points (2010 and 2012). This Indian study by Subramanian *et al*, utilized a household-based sampling strategy for the location of mosquito traps and concluded that the strategy produced reproducible results which were in line with the observed LF infection trends found in the human population [32]. Our study used a similar design, observed similar findings and equally led to same conclusions. In addition, our mosquito survey findings corroborate results of a population-based survey that was done in parallel [20]. In Samoa, a significant reduction of LF infection prevalence in mosquito vectors was also observed between 2018 and 2019 time-points following treatment with one round of IDA triple-therapy. However, contrary to our study where both mosquito and human population-based surveys showed a similar trend, in the Samoa study there was no significant change in human Ag prevalence [33].

In our study, only one individual was found to be Mf positive during baseline survey. Coincidentally, this Mf positivity was found in the village (Mreroni) that had the highest overall prevalence of filarial DNA in the baseline survey indicating concurrence between the two LF infection signals. This finding agrees with a study by Howlett *et al* that showed a strong correlation between PCR-positive mosquitoes and Mf in humans at primary sampling units and households thus demonstrating MX utility as an indicator for LF prevalence and disease endpoint in an area [34].

On the other hand, two nighttime DBS samples tested positive for filarial DNA by qPCR at IIS-1 despite the night blood samples from the respective individuals being negative for Mf. This could be a pointer to the fact that qPCR is much more sensitive compared to microscopy-based Mf testing in humans and might be able to detect ultra-low microfilaremia and thus provide a real-time potential of ongoing transmission compared to CFA [18,26,35,36]. This also exemplifies the utility of MX in detection of Mf missed by the human population-based surveys. However, while this is effectively an indication of the presence of Mf in the blood, currently there are no express guidelines for the use of night blood Mf positivity by qPCR for programmatic decision making. Nonetheless, if the two nighttime DBS qPCR positive samples were to be interpreted as Mf positive, they would still be below 1% microfilaremia threshold for safe stopping of the MDA [15]. This positivity should be treated with caution since a positive result by MX cannot necessarily be taken to be an indicator of ongoing LF transmission noting that the testing is not specifically targeted to the infective L3 stage of *W. bancrofti*. Such a finding should however underscore the need for a continued surveillance to detect and mitigate residual infection, and timely arrest of any recrudescence [37,38].

In terms of prevalence by mosquito genus, *Culex* mosquito had the highest prevalence of 0.26% at baseline that significantly reduced to 0.06% at IIS-1. This result indicates that the prevalence by *Culex* at IIS-1 was below the provisional positivity threshold for stopping MDA by MX of < 0.25% based on the current guidelines [39]. Since similar results were arrived at using CFA in the human survey, our findings therefore provide additional evidence for possible interruption of LF in the two EUs(20). Further, the agreement between mosquito and human positivity suggests that MX can be used a complementary epidemiological indicator of the LF infection levels in these areas.

Interestingly, unlike in Jomvu, in Lamu there was a slight increase in prevalence by genus *Culex* from 0% at baseline to 0.01% at IIS-1 survey in overall prevalence. However, this increase would not necessarily be indicative of any meaningful resurgence of the infection but rather may be due to natural stochastic variation from the sampling strategy. However, the fact that Ndau village in Lamu had 8 CFA positive individuals and a MX positive pool at IIS-1 and the results being consistent with previous study done in human in 2015, would be a vital evidence for the village to be treated as a hotspot thus warranting targeted surveillance in future [13].

Our findings indicate that Gravid traps, which target the genus *Culex* mosquito, had the highest yield of the mosquitoes. This was not entirely unexpected since this genus has historically been associated with transmission of the LF infection as the main vector both in Kenya and in the larger East Africa region [40–42]. In an entomological survey done in Central Nepal, it was also shown that Gravid traps were the most efficient trap for the collection of mosquitoes in LF endemic areas, particularly in areas where *Culex* is the principal vector for LF [27,36,43]. The ability to trap large numbers of mosquitoes, the high sensitivity of qPCR to detect filarial DNA, and the finding that mosquito pool positivity can be used as a proxy indicator of the infection in humans provides support for the use of xenomonitoring as a post-MDA surveillance tool for LF elimination programmes.

In terms of the EU, Jomvu had comparatively higher number of the *Culex* and filarial DNA prevalence compared to Lamu. This could be a result of the fact that Jomvu is an urban and peri-urban setting, an environment that would naturally favour *Culex* mosquito's dominance [1,42,44,45]. In any case, Mombasa City has previously been observed to have large numbers of the *C. quinquefasciatus* mosquitoes despite showing low prevalence of Mf [46,47]. Although no Genus *Anopheles* pool was found to be positive for filarial DNA in our study, the transmission of the LF infection by the different species of the *Anopheles* genus in the inland part of the Kenya and in Tanzania has previously also been documented [2,48,49]. In the West African region, genus *Anopheles* has previously also been shown to be an important vector in Togo [50].

This notwithstanding, this study experienced challenges in getting sufficient numbers of *Anopheles* mosquitoes. Genus *Anopheles* is naturally known to require fresh water for breeding and thus makes its yields, densities and spatial distribution subject to seasonal variations [51]. This could be a major factor that could have led to low collections during the current study. General reductions in mosquito yield that is attributable to dry season was also evident in a study that was conducted in Samoa [33]. Lately, urbanization and climate change are thought to be driving changes in human

populations, as well as in the natural habitat and microclimate in sub-Saharan Africa, thus affecting vector distribution in an area [52]. These observations, serve as useful hints and guides for programme implementers to consider in order to achieve large numbers of mosquito collection. In addition, it would be imperative for future studies to devise mosquito collections strategies that would overcome these challenges and improve on yields during post validation surveillance (PVS) surveys. This will enhance the dependability for this genus *Anopheles* in the monitoring of LF infection in the two areas and in other regions.

This study mainly targeted *Culex* and *Anopheles* as they are the main vectors responsible for the transmission of LF in rural Kenya [42,47–49]. However, increased urbanization trends in the major towns along the Kenyan coast in the recent past have been associated with rapid increase in *Aedes* mosquito species and this has been suspected to be the main driver of the occasional outbreaks of arboviral diseases in the region [53,54]. Also, the widespread practice of storage of fresh water in open containers as result of erratic water supply in the urban settings has provided suitable conditions for the breeding of the *Aedes* species [55]. Therefore, *Aedes* mosquitoes were collected at baseline to establish their role in the transmission of LF in the study area. However, all the *Aedes* mosquito pools tested at baseline were negative signifying a limited role in LF transmission and thus in line with available LF transmission literature also, they were not collected at IIS-1 [1]. The collection of the three species could, however, be considered in the future as this will provide a window of opportunity for an integrated platform for monitoring LF and other mosquito borne co-endemic infections including malaria and arboviruses in the area with minimal implications on logistics, cost and time in the field.

In our study, both the random and the purposive clusters showed largely similar results suggesting that they may be used to provide similar epidemiological results thus no need for such classifications during sampling of the villages. However, targeted individual (purposive) villages for the evaluation of the existence of any residual infection (hotpots) have previously been applied as part of larger epidemiological assessments, although the sampling strategies and sample sizes are typically powered to enable decision making at EU levels, not at village level [35,56]. In line with this principle, some villages in our study were observed to have relatively high prevalence for filarial DNA in comparison to WHO guidelines and these villages would therefore be considered as possible hotspots requiring special attention during surveillance in future.

### 4.1. Study strengths and weaknesses

The primary strength of the current study is that is has provided an opportunity to compare circulating filarial antigen, microfilariae, and filarial DNA infection indicators for monitoring LF elimination efforts and more so under triple therapy treatment context. This study was also undertaken in two diverse settings characterized by the rural-urban settings, islands versus mainland, and hard-to-reach communities, thus making the findings more generalizable across other endemic settings. Additionally, having both baseline and IIS-1 timepoints enabled incorporation of the useful lessons learnt at baseline to IIS-1 and provided a chance to generate data that will inform further programmatic actions, including Kenyan dossier development and designing post validation surveillance (PVS) activities. However, the current study was hampered by the Covid 19 outbreak in 2020 which posed significant impact on the costs associated to disease mitigations, logistical and implementational challenges due to Covid-related community stigma.

### 5. Conclusion

This study demonstrates that LF transmission has likely been interrupted in Lamu County and Jomvu sub-county, Mombasa County in the Kenyan Coast thus warranting stoppage of the MDA. These findings are corroborated by the human results and provide additional evidence that IDA triple-drug therapy regimen can be used in accelerating elimination of the LF infection in areas that have been grappling with sub-optimal performance of two drug regimen of diethylcarbamazine and albendazole (DA) treatment. The study also affirms that *Culex* is the dominant LF vector in Kenya and Gravid traps are the most suitable trap for use in the collection of the mosquitoes. Finally, this study shows that MX can detect Mf that

are not detectable through human population-based surveys thus making it a suitable, robust and alternative LF infection monitoring tool with minimal intrusion to human lives. Additionally, MX is also deemed to have an extra advantage in that it is indicative of current active infection (not necessarily ongoing transmission) and a platform that is potentially amenable to the establishment of an integrated surveillance of other co-endemic mosquito borne infections in the spirit of one-health approach. Further research could be considered on the cost implications of the technique and the feasibility for scalability for wide scale surveillance of the infection(s) by the NTD programs.

## Acknowledgments

Our special regards go to Vector Borne and Neglected Tropical Diseases Unit (VBNTDU), Ministry of Health, Kenya and the County departments of health in Lamu and Mombasa counties for their unwavering support during the implementation of the study. Special thanks go to Dr. Patrick Lammie of the Neglected Tropical Diseases Support Center (NTDSC) at Atlanta, Georgia, USA and Prof. Nils Pilotte of Cincinnati University. Dr. Angus McLure of The Australian National University is thanked for providing the PoolTools v0.1.4 software used to calculate filarial DNA prevalence in this study. We acknowledge the enormous support by the co-investigators, the entire study team and local personnel for their efforts in implementation of the study.

This paper is published with the permission of the Director General, KEMRI.

## Author contributions

**Conceptualization:** Henry Kanyi, David Odongo, Walter Jaoko, Katherine Gass, Sammy M. Njenga.

**Data curation:** Henry Kanyi, Collins Okoyo, Sammy M. Njenga.

**Formal analysis:** Henry Kanyi, Benard Chieng, Sylvie Araka, Collins Okoyo.

**Investigation:** Katherine Gass, Sammy M. Njenga.

**Methodology:** Henry Kanyi, David Odongo, Walter Jaoko, Katherine Gass, Benard Chieng, Sylvie Araka, Sammy M. Njenga.

**Project administration:** Henry Kanyi, Sammy M. Njenga.

**Software:** Collins Okoyo.

**Supervision:** Henry Kanyi, David Odongo, Walter Jaoko, Katherine Gass, Sammy M. Njenga.

**Validation:** Henry Kanyi, Sammy M. Njenga.

**Visualization:** Katherine Gass, Collins Okoyo, Sammy M. Njenga.

**Writing – original draft:** Henry Kanyi, David Odongo, Walter Jaoko, Katherine Gass, Benard Chieng, Sylvie Araka, Collins Okoyo, Sammy M. Njenga.

**Writing – review & editing:** Henry Kanyi, David Odongo, Walter Jaoko, Katherine Gass, Collins Okoyo, Sammy M. Njenga.

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
