## [Decision Letter · Decision Letter 0]

3 Sep 2025

PGPH-D-25-01565

The role of molecular xenomonitoring in the assessment of the impact of Ivermectin, Diethylcarbamazine Citrate and Albendazole triple-drug therapy treatment to the lymphatic filariasis infection levels in the Kenyan coast.

Dear Dr. Kanyi,

Thank you for submitting your manuscript to PLOS Global Public Health. After careful consideration, we feel that it has merit but does not fully meet PLOS Global Public Health’s publication criteria as it currently stands. Therefore, we invite you to submit a revised version of the manuscript that addresses the points raised during the review process.

I would like to sincerely apologise for the delay you have incurred with your submission. It has been exceptionally difficult to secure reviewers to evaluate your study. We have now received five completed reviews; the comments are available below. Some reviewers have raised significant scientific concerns about the study that need to be addressed in a revision.

Please revise the manuscript to address all the reviewer's comments in a point-by-point response in order to ensure it is meeting the journal's publication criteria. Please note that the revised manuscript will need to undergo further review, we thus cannot at this point anticipate the outcome of the evaluation process.

We look forward to receiving your revised manuscript.

Kind regards,

Miquel Vall-llosera Camps, Ph.D.

Staff Editor

Journal Requirements:

Reviewers' comments:

Reviewer's Responses to Questions

**Comments to the Author**

1. Does this manuscript meet PLOS Global Public Health’s publication criteria?

Reviewer #1: Yes

Reviewer #2: Yes

Reviewer #3: Partly

Reviewer #4: Yes

Reviewer #5: Yes

2. Has the statistical analysis been performed appropriately and rigorously?

Reviewer #1: Yes

Reviewer #2: Yes

Reviewer #3: No

Reviewer #4: Yes

Reviewer #5: No

3. Have the authors made all data underlying the findings in their manuscript fully available (please refer to the Data Availability Statement at the start of the manuscript PDF file)?

Reviewer #1: Yes

Reviewer #2: Yes

Reviewer #3: Yes

Reviewer #4: Yes

Reviewer #5: No

4. Is the manuscript presented in an intelligible fashion and written in standard English?

Reviewer #1: Yes

Reviewer #2: Yes

Reviewer #3: Yes

Reviewer #4: Yes

Reviewer #5: Yes

Reviewer #1: Dear Author,

Kindly go through the following minor corrections.

Thank you

GENERAL:

There are a few minor typographical issues (e.g., inconsistent spacing before and after punctuation and citations) that should be addressed during final proofreading. Still, they do not detract from the overall clarity of the manuscript. I recommend a thorough grammar and style review to ensure the manuscript meets publication standards.

RESULT:

"during baseline assessment" is repeated in lines 265–266.

The dominance of the Culex genus is restated multiple times in similar ways; it can be made more concise.

Use consistent formatting for genus names (italicize Anopheles, Aedes, Culex).

It’s unclear what "IIS-1" refers to for readers who haven’t seen the acronym defined recently. Consider writing it out or reminding the reader.

The tables are generally informative and support the results well. However, a few tables appear overly condensed or too small, making the data difficult to read or interpret clearly. Consider increasing font size and spacing for proper reading. If feasible, split very dense tables into multiple simpler ones.

Thank you.

Reviewer #2: This manuscript presents timely and important research evaluating the impact of triple-drug IDA mass drug administration on lymphatic filariasis (LF) using molecular xenomonitoring (MX) in coastal Kenya. The study offers valuable insights that are highly relevant for LF elimination programs and supports the operational value of MX as a surveillance tool. The integration of entomological data with human infection prevalence strengthens the quality of the findings.

Strengths:

The study design, combining human and vector-based data, is innovative and fits well within global efforts to enhance NTD surveillance.

The laboratory protocols, including pooling strategies and qPCR methods, are clearly described and adhere to high scientific standards.

Ethical approvals and informed consent procedures are adequately documented.

The manuscript demonstrates a commitment to open data, in line with PLOS policies.

Major Areas for Improvement:

Title Simplification

The current title is excessively long and includes redundant phrasing ("therapy treatment"). Consider shortening to enhance clarity and reader engagement.

Sampling Strategy and Household Coverage

While the study mentions the use of both purposive and random villages, it does not explain the rationale for these selections. Please clarify why these methods were used and their implications for the representativeness of the findings.

There is no information on household coverage; please specify how many households were sampled at each site and whether the same ones were revisited in follow-up surveys.

Statistical Reporting

Some of the reported prevalence changes lack p-values or confidence intervals. Please provide statistical tests for key comparisons to support the strength of your conclusions.

Explain clearly how percentage reductions (e.g., “76.9% decrease”) were calculated and whether they are statistically significant.

Programmatic Relevance

The discussion could be expanded to provide practical guidance for national LF programs:

How can MX be integrated into routine surveillance?

What are the training, cost, and scalability considerations?

How could MX be leveraged for integrated vector-borne disease surveillance under a One Health approach?

Minor Issues to Address:

Ensure scientific names (e.g., Wuchereria bancrofti) are consistently italicized.

Expand figure legends so that each is interpretable independently of the text.

Some sentences (e.g., Lines 411–414) are overly long or awkward; consider editing for clarity and fluency.

Conclusion:

This is a well-executed and timely study that makes a significant contribution to the literature on LF elimination. I recommend major revisions to clarify the sampling strategy, reinforce statistical reporting, and expand the discussion of programmatic applications. With these refinements, the manuscript will be a strong fit for PLOS Global Public Health

Reviewer #3: OVERALL:

This study provides some insight into the impact of a triple drug combination therapy on lymphatic filariasis infection in Kenya. Grammar and phrasing needs to be reviewed in the editing phases. The writing can be more concise and the authors should consider removing words such as "probably", "not necessarily" and other adverbials.

INTRODUCTION:

- The author immediately introduces Kenya and then discusses the Pacific endemic islands. It is not clear why. Where is FL most prevalent? How many people are affected? How big is the problem in Kenya?

- In lines 97-101, the two study sites are first mentioned. Is there a reason why these study sites were chosen? Is there any reference to support this? Are lines 99-101 meant to describe the purpose of the current study?

METHODS:

- The manuscript does not have a detailed purpose and explanation of the statistical methods used in the methods section. The spatial differences (random vs. purposive) and temporal differences (baseline vs. IIS-1) could be made much clearer. It was unclear to me why there were "random" and "purposive" clusters, the meaning of these groups were not outlined in the methods. Please outline these clearly for the reader in the methods section. Aside from this, I can see that the statistics used were relevant to the nature of the work.

RESULTS:

- Line 268 describes “the collection”. Can the author elaborate on what this refers to: the collection of mosquito samples or human related samples?

- Section 3.1 is said to describe the demographic characteristics, however it outlines the sample collection for the mosquitoes. Consider rephrasing the title for this section.

- Table 2 shows a very low prevalence of LF. I am uncertain how prevalence was calculated here.

- E.g. For “all sites” in Jomvu at baseline: If number of positive pools / number of pools analysed = 30 / 323 = 0.0929. Where does 0.40 (0.27-0.56) come from?

- For table 2, Consider adding that the samples were obtained from mosquitoes.

- As the tables go on, I am still uncertain about the calculations of prevalence. Please specify how these were calculated.

DISCUSSION:

- It would be more impactful if the first paragraph (lines 368-381 only discussed the novel findings from this paper, before going into how this paper relates to the remainder of the literature.

- Lines 476-479 discuss how the study sites allowed for generalisable conclusions to be made. This is a bold point given that there are many epidemiologies associated with disease status. Please elaborate on this by providing more context on these study sites in the methods and how they are similar epidemiologically to other settings where LF is prevalent.

- The conclusion "This study demonstrates that LF transmission has likely been interrupted in Lamu County and

Jomvu sub-county, Mombasa County in the Kenyan Coast thus warranting stoppage of the MDA." is

Reviewer #4: Thank you for giving chance to review this very impactful and deserved work. The data availability found supporting the conclusions and the research thesis incorporated research ethical principles. The recommendation and conclusion was relevant and draw based the authors finding.

Reviewer #5: Thank you for the opportunity to review your manuscript, “The role of molecular xenomonitoring in the assessment of the impact of Ivermectin, Diethylcarbamazine Citrate and Albendazole triple-drug therapy treatment to the lymphatic filariasis infection levels in the Kenyan coast.” This is an important study that provides valuable data on the impact of IDA therapy and the utility of molecular xenomonitoring in the Kenyan context. The work has clear programmatic relevance. To improve the manuscript for potential publication, I offer the following comments for your consideration.

Major Comments

1. Methodological Clarity and Study Design

The Methods section requires significant revision to provide the clarity and detail necessary for readers to fully understand and evaluate the study.

Study Design (Line 139): The description “repeated cross-sectional entomological surveys” is ambiguous. Please clarify if this was a longitudinal study with a pre- and post-intervention assessment of the same areas, or two independent cross-sectional surveys. A clear, standard term for the study design should be used consistently.

Sampling Strategy (Line 141): The use of a mixed sampling strategy (“random” and “purposive”) needs justification. Please explain the rationale for selecting some villages purposively and describe the criteria used for this selection.

Study Population: The manuscript lacks a dedicated "Study Population" section. Information is currently scattered. Please create a distinct section that clearly defines the human and mosquito populations studied, including explicit inclusion and exclusion criteria for both. Details on household numbers (lines 261-262) should be moved from the Results to this new section in the Methods.

Study Setting (Line 144-150): Please provide more context.

Justify why these specific sites were chosen (e.g., known high LF burden, representing diverse ecological settings, etc.).

Provide essential details about the IDA mass drug administration (MDA) itself: When, where, and by whom was the therapy administered? This context is crucial for understanding the intervention whose impact is being assessed.

Considering the influence of climate on vector populations, adding brief details on the local environmental conditions (e.g., rainy/dry seasons) would enrich the site description.

Changes in Methodology: The manuscript notes several changes between the baseline and follow-up surveys. These must be justified within the Methods section.

Age Brackets (Line 153): Explain the reason for changing the age criteria for the older population from ≥10 years at baseline to ≥18 years at IIS-1. Please also provide a specific upper age limit or range for the ≥18 group.

Mosquito Traps (Line 176): State in the Methods why the BG-sentinel trap was dropped for the IIS-1 survey.

2. Ethical Considerations (Lines 131-137)

While the ethics statement is strong, it can be enhanced for greater clarity and completeness.

Consent Process: The phrase “study participants and household heads provided a written informed consent” could be ambiguous. To emphasize individual autonomy, please clarify that individual written informed consent was obtained from all competent adult participants, and that permission from household heads was sought in line with cultural norms, not as a substitute for individual consent.

Assent for Minors: If children or adolescents under 18 were included, the process for obtaining their assent, in addition to parental/guardian consent, must be explicitly described.

Sample Storage: The consent for “further molecular analysis” is noted, but please specify if this included consent for long-term storage of samples and for what duration.

Suggested Revision: Please consider incorporating elements from the following example to improve clarity: “The study received ethical approval from the Scientific and Ethics Review Unit (SERU) of the Kenya Medical Research Institute (KEMRI) (KEMRI/SERU protocol No. 4523). Permission to conduct the study and collect human and mosquito samples was further obtained from the respective county governments. Community mobilization and sensitization were conducted through meetings with local leaders and community members at the village level. All adult participants provided individual written informed consent. For participants under 18, written parental permission was obtained, along with written assent from the minor where appropriate. The consent process, conducted in the local language, included provisions for non-literate participants and specified that samples could be stored for future molecular analysis related to vector-borne disease research.”

3. Reporting and Analysis of Results

Pool Positivity vs. Estimated Prevalence (Abstract, Lines 34-38): There is a significant discrepancy between the reported percentages (0.26% and 0.06%) and the raw pool positivity rates that can be calculated from the data provided (30/486 = 6.17% and 22/1695 = 1.30%). Please clarify in the text that the reported percentages are the estimated DNA prevalence from PoolTools and are distinct from the raw pool positivity rate. This is a critical point for reader comprehension.

Measures of Impact (Tables 2-4): The study title implies an assessment of "impact." While the descriptive results are valuable, the analysis would be strengthened by reporting formal measures of impact beyond relative reduction. Please consider including measures such as Absolute Risk Reduction (ARR) or Risk Difference to provide a more complete picture of the intervention's effect.

Table Structure: The structure of the tables, particularly the reporting of 'n', can be confusing when trying to distinguish between the number of mosquitoes and the number of pools. Please refine the table legends and column headers for clarity.

Minor Comments

Title: The current title is long, contains some redundancy ("therapy treatment"), and could give the impression of a clinical trial. Please consider revising it to better reflect the observational study design and primary objective. For instance: “Impact of Ivermectin, Diethylcarbamazine Citrate and Albendazole Therapy on Lymphatic Filariasis in Coastal Kenya: A Longitudinal Assessment Using Molecular Xenomonitoring.”

Abstract (Line 22): Defining Diethylcarbamazine as "(DEC)" immediately before defining the three-drug combination as "(IDA)" is confusing. Since DEC is not referred to again on its own, consider removing this abbreviation.

Introduction (Lines 63-78): Several foundational statements about LF and the WHO's GPELF are presented without citations. While some may be considered common knowledge, it is best practice to reference key facts and programmatic statements. Please add appropriate references.

Formatting (Line 144): Please ensure consistent formatting for citations (e.g., space before parenthesis, period after "al.").

Terminology (Line 158): Please confirm that "Mf" is the standard, accepted abbreviation for microfilariae in the context of this journal.

Thank you for your work on this important topic. I believe that addressing these points will significantly improve the manuscript's clarity, rigor, and impact. Please find the detailed line by line comments attached.

**Do you want your identity to be public for this peer review?** For information about this choice, including consent withdrawal, please see our Privacy Policy

Reviewer #1: No

Reviewer #2: **Yes: ** Hodabalo AWIZOBA

Reviewer #3: No

Reviewer #4: **Yes: ** Abebe Nigussie Ayele, Researcher and Lecturer at Debre Berhan University, Asrat Woldeyes Health Science Campus ,

Reviewer #5: **Yes: ** Suleiman Idris Ahmad

---

## [Decision Letter · Decision Letter 1]

23 Nov 2025

The role of molecular xenomonitoring for assessing the impact of Ivermectin, Diethylcarbamazine Citrate and Albendazole triple-drug treatment against lymphatic filariasis: A cross-sectional study in the Kenyan coastal region

PGPH-D-25-01565R1

Dear Mr Kanyi,

We are pleased to inform you that your manuscript 'The role of molecular xenomonitoring for assessing the impact of Ivermectin, Diethylcarbamazine Citrate and Albendazole triple-drug treatment against lymphatic filariasis: A cross-sectional study in the Kenyan coastal region' has been provisionally accepted for publication in PLOS Global Public Health.

Best regards,

Julia Robinson

Executive Editor

Reviewer Comments (if any, and for reference):

Reviewer's Responses to Questions

**Comments to the Author**

Reviewer #1: All comments have been addressed

Reviewer #2: All comments have been addressed

Reviewer #3: All comments have been addressed

Reviewer #4: All comments have been addressed

publication criteria?

Reviewer #1: Yes

Reviewer #2: Yes

Reviewer #3: Yes

Reviewer #4: Yes

3. Has the statistical analysis been performed appropriately and rigorously?

Reviewer #1: Yes

Reviewer #2: Yes

Reviewer #3: Yes

Reviewer #4: Yes

4. Have the authors made all data underlying the findings in their manuscript fully available (please refer to the Data Availability Statement at the start of the manuscript PDF file)?

Reviewer #1: Yes

Reviewer #2: Yes

Reviewer #3: Yes

Reviewer #4: Yes

5. Is the manuscript presented in an intelligible fashion and written in standard English?

Reviewer #1: Yes

Reviewer #2: Yes

Reviewer #3: Yes

Reviewer #4: Yes

Reviewer #1: Dear Author,

Thank you for your submission. The manuscript is well-organized, informative, and clearly presents the study objectives, methods, results, and conclusions. All major concerns from the previous review appear to have been addressed.

Before final submission, I recommend ensuring the following:

Verify that all scientific names (e.g., Wuchereria bancrofti) are italicized and consistently formatted.

Double-check that citations and references, if any, are accurate and formatted according to the journal’s guidelines.

Confirm that the links or datasets referenced are accessible.

Consider clarifying a few phrasing redundancies (e.g., “accelerate LF elimination of LF”) for smoother readability.

Ensure consistent use of abbreviations (define once, then use the acronym thereafter).

Check numerical data (percentages and sample sizes) for consistency between text and any accompanying tables or figures.

Overall, excellent work.

Best regards,

Reviewer #2: This is an excellent and timely manuscript that makes a valuable contribution to the field of lymphatic filariasis (LF) elimination. The study is well designed, methodologically sound, and clearly demonstrates the effectiveness of the triple-drug (IDA) therapy and the usefulness of molecular xenomonitoring (MX) as a sensitive surveillance tool.

The integration of entomological and human data is a major strength, and the findings are highly relevant for national and global LF elimination programs. Ethical standards and data transparency are well maintained.

Only very minor editorial improvements (grammar polishing and clearer figure legends) are suggested, but these do not affect the scientific quality of the work.

Reviewer #3: This manuscript has been revised in a manner that meets the criteria for a clear and interesting scientific paper. The manuscript is technically sound with clarified statistical analyses and methods that are appropriate. The authors have explained their data that support their findings very well, and have revised the grammatical errors from the first version.

Reviewer #4: I am lucky to review this innovative and impactful paper. The authors are better research writing skills and commitment to publish this article to highly indexed and impactful journals. thank you.

**Do you want your identity to be public for this peer review?** For information about this choice, including consent withdrawal, please see our Privacy Policy

Reviewer #1: No

Reviewer #2: **Yes: ** Hodabalo AWIZOBA

Reviewer #3: No

Reviewer #4: No
